# Genome-Wide Identification and Expression Patterns of the F-box Family in Poplar under Salt Stress

**DOI:** 10.3390/ijms231810934

**Published:** 2022-09-18

**Authors:** Gaofeng Fan, Xinhui Xia, Wenjing Yao, Zihan Cheng, Xuemei Zhang, Jiahui Jiang, Boru Zhou, Tingbo Jiang

**Affiliations:** 1State Key Laboratory of Tree Genetics and Breeding, Northeast Forestry University, Harbin 150040, China; 2Bamboo Research Institute, Nanjing Forestry University, 159 Longpan Road, Nanjing 210037, China

**Keywords:** poplar, F-box family, phylogenetic analysis, expression pattern, co-expression network

## Abstract

The F-box family exists in a wide variety of plants and plays an extremely important role in plant growth, development and stress responses. However, systematic studies of F-box family have not been reported in *populus trichocarpa*. In the present study, 245 *PtrFBX* proteins in total were identified, and a phylogenetic tree was constructed on the basis of their C-terminal conserved domains, which was divided into 16 groups (A–P). F-box proteins were located in 19 chromosomes and six scaffolds, and segmental duplication was main force for the evolution of the F-box family in poplar. Collinearity analysis was conducted between poplar and other species including *Arabidopsis thaliana*, *Glycine max*, *Anemone vitifolia* Buch, *Oryza sativa* and *Zea mays*, which indicated that poplar has a relatively close relationship with *G. max*. The promoter regions of *PtrFBX* genes mainly contain two kinds of *cis*-elements, including hormone-responsive elements and stress-related elements. Transcriptome analysis indicated that there were 82 differentially expressed *PtrFBX* genes (DEGs), among which 64 DEGs were in the roots, 17 in the leaves and 26 in the stems. In addition, a co-expression network analysis of four representative *PtrFBX* genes indicated that their co-expression gene sets were mainly involved in abiotic stress responses and complex physiological processes. Using bioinformatic methods, we explored the structure, evolution and expression pattern of F-box genes in poplar, which provided clues to the molecular function of F-box family members and the screening of salt-tolerant *PtrFBX* genes.

## 1. Introduction

Plants are subjected to a variety of challenges, including drought, salt, high or low temperature, heavy metals and so on. In particular, salt is one of the biggest threats, leading to soil salinization of 19.5% of agricultural land every year, and it is expected that 30% of agricultural land will be salinized by 2028 [1] and more than 50% will be affected by salinization by 2050 [2]. So far, the genetic engineering technology is regarded as one of the most reliable and effective methods of developing salt-tolerant plants [3].

The ubiquitin/26S proteasome system (UPS) is an extremely important intracellular regulatory network to maintain the normal life activities of plants after receiving external stimulus signals. It achieves the purpose of pre-transcriptional regulation by selectively recognizing specific substrates and degrading the target proteins [4]. It involves three consecutive steps: (1) activating ubiquitin with the E1 enzyme, (2) transferring the activated ubiquitin to the E2 enzyme and (3) transporting ubiquitin to the E3 enzyme to degrade the target proteins. The SCF complexes are composed of SKP1, CUL1 and the F-box proteins that are main activators of E3 ligase [5]. As one component of the SCF complex, the F-box members are responsible for specific recognition and degradation of the target proteins, which may take part in different biological pathways [6].

The F-box family is named from its conservative F-box domain, which was firstly found in human Cyclin F [7]. A great many F-box genes have been found in higher plants. For example, there were 725, 359, 517 and 226 F-box proteins identified in *G. max*, *Z. mays*, *Malus domestica* and *Pyrus* spp., respectively [8,9,10,11]. The conserved domain of F-box proteins is the F-box with about 60 aa at the N-terminal, which interacts with SKP1 and is indispensable for protein–protein interaction [12]. The C-terminal contains different conserved domains including FBA (which may be involved in the binding of target ubiquitin), FBD (which is involved in nuclear processes), Tub (which may be associated with obesity, blindness and deafness), Kelch (which may regulate the processes of proliferation and/or differentiation), WD40, DUF295, PP2, actin, FIST, ARM, PAS, Sel1, Herpes [13,14,15,16]. Furthermore, F-box genes with the Kelch domain can be named the FBK subgroup; likewise, the FBL subgroup with the LRR domain, FBD with the FBD domain, and FBO with other domains (FIST, PAS, actin, ARM, etc.) [17].

F-box proteins are vital for transcription regulation [18]. Studies have reported that F-box proteins play an important role in regulating flowering, root growth, seed dormancy and germination, leaf senescence, hormone signal transduction (IAA, GA, ABA, JA, ethylene) and stress responses in plants [17]. *UFO* was the first F-box gene identified from *Arabidopsis thaliana*, whose abnormal expression caused the abnormal development of flower organs [19]. *MAX2*, which encoded an F-box gene with an LRR domain, reduced water loss and changed the osmotic pressure to enhance ABA sensitivity in *Arabidopsis* [20]. *AFB 1–3*, a F-box protein from *Arabidopsis*, could bind Aux/IAA protein to promote the auxin response [21]. *TaPUB26*, a negatively regulated F-box factor, was able to change the Na^+^/K^+^ contents and affect the salt sensitivity of transgenic *Brachypodium distachyon* [22]. In a similar way, *TaFBA1* and *ATPP2-B11* could influence the salt tolerance of plants [23,24]. However, studies on F-box proteins are limited in poplar.

With the rapid development of high-throughput sequencing, genome-wide sequencing has been completed in many plant species such as *Arabidopsis*, soybean, cotton, rice and *populus trichocarpa* [25,26,27,28,29], and provides great convenience in studies of the evolutionary relationships among different plant species. In this study, we comprehensively identified F-box family from poplar, analyzed their protein structures and motifs, and explored their driving force in poplar’s evolution. The upstream promoter regions of the F-box genes were also investigated, which contained many *cis*-acting elements related to hormone and stress responses. In addition, the expression patterns of F-box genes in the roots, stems and leaves of poplar were explored under salt stress conditions. Finally, four interesting genes were selected for co-expression network analysis to further excavate their diverse biological functions. To date, this study is the first comprehensive report of F-box family in poplar, which may provide some insights for the screening of stress-related F-box genes.

## 2. Results

### 2.1. Genomic-Wide Identification and GO Annotation of the F-box Family in Poplar

F-box proteins with the representative HMM seeds from Pfam database were obtained from the Phytozome database. The conserved F-box domain was located at the N-terminal of the F-box proteins in poplar. All candidate proteins were screened by the SMART and Pfam databases for removing the proteins without an F-box domain and all redundant proteins. Finally, 245 members with the F-box domain were identified and used for our subsequent analysis (Appendix A). According to their location on the poplar chromosomes, the 245 genes were numbered as *PtrFBX*1–*PtrFBX*245.

In order to understand the biological pathway involved in the F-box family in poplar, the functions of the 245 F-box proteins were predicted by EggNOG database. The results showed that most F-box genes played crucial roles in metabolic processes (46) and biological regulation (38) (Appendix A). Other F-box genes were involved in many essential biological processes such as the response to stimulus, various signaling pathways, growth and development, protein transport and protein modification. The Benjamin–Hochberg method was used to verify some important GO enrichment contents, including the cellular responses to hormone stimuli (GO:0032870), endogenous stimuli (GO:0071495) and auxin stimuli (GO:0071365); hormone-mediated signaling pathways (GO:0009755); responses to fungi (GO:0009620); pollen development (GO:0009555); multicellular organismal reproductive processes (GO:0048609); protein modification processes (GO:0036211) and the transferase complex (GO:1990234).

### 2.2. Analysis of Amino Acid Residues and Conserved Domains of F-box Proteins in Poplar

F-box domain is an indispensable part of F-box family. The amino acid residues of F-box domain are shown in Figure 1a. We found the sites 7, 8, 18, 22, 41 and 45 are highly conserved in the 245 F-box proteins, which are Leu (L, 71.4%), Pro (P, 75.5%), Ile (I, 78.4%), leu (L, 73.9%), Val (V, 76.7%) and Trp (T, 80.4%), respectively. These mostly conserved residues are consistent with former studies [9,30], suggesting that these residues have maintained evolutionary stability in many plants, which may contribute to specific binding sites for the formation of SCF complexes in SKP proteins. According to the conserved domains at the C-terminal, the F-box proteins could be further divided into diverse subgroups including FBA, FBK, FBU, FBT, FBF, FBT, FBL, FBP, FBW and FBO (Appendix A). Most F-box proteins (135) only have one F-box domain at the N-terminal, following an unknown domain at the C-terminal, which belongs to the FBU subgroup. Other subgroups such as FBA (28), FBK (20), FBL (14), FBD (12), FBT (11), FBF (3), FBP (2) and FBW (1) have a specific domain including FBA, Kelch, LRR, FBD, TUB, DUF295, PP2 and WD40, respectively. FBO (19) has several domains such as PAS-Kelch, FIST, Actin, LysM, Sel1, Herpes, ARM, Cupin-like and FAE1/Type III polyketide synthase-like protein (Figure 1b).

### 2.3. Phylogenetic Tree and Structure Analysis of the F-box Proteins in Poplar

In order to understand the evolutionary relationships of the F-box family in poplar, a phylogenetic tree with 245 full-length protein sequences was constructed by the neighbor-joining (NJ) method (Figure 2). All proteins were divided into Groups A–P. Among these, Group O mainly contains the FBA and FBU subgroups, Group A is mainly composed of the FBU and FBD subgroups, Group L is mainly composed of the FBU subgroup, while Groups E, M and P are the smallest subgroups with only four members. Furthermore, we analyzed the intron/exon structures and motifs of all proteins (Figure 3). Each group shared similar structures and motifs. For example, 18 out of 28 members belonging to Group O have no intron and only one exon. Most members of Groups D and F have similar structures, that is, one exon and no intron. In addition, 20 conserved motifs in total were found in the *PtrFBX* proteins by MEME, and these motifs were further annotated by the SMART and Pfam databases (Appendix A). Motifs 2 and 19 were identified to be part of the F-box domain, and Motifs 3, 4, 7, 13 and 18 were regarded as the Tub domain. Motif 9 is the Kelch domain, Motif 12 is the FBA domain, and the rest are not annotated. The prediction showed that *PtrFBX*6, *PtrFBX*62, *PtrFBX*63 and *PtrFBX*171 did not have Motif 2, and all the others had one Motif 2, except *PtrFBX*220 had two, while Motif 15 exists in Groups A, B, C and D. As many as 11 members of Group D have Motif 5, and Motif 7 only exists in Groups M and L. Interestingly, Motifs 3, 4, 13, 16 and 18 were found only in Group H, and no recognizable motif was found in the *PtrFBX*102 protein.

### 2.4. Chromosomal Location and Duplication Events of the F-box Family in Poplar

As shown in Figure 4a, all *PtrFBX* genes were distributed on all 19 chromosomes and six scaffolds. The largest number (32) of genes was on Chr1, followed by 20 genes on chromosome 6 and 8. There are 6, 8, 7 and 3 genes on Chr12, 15, 16 and 19, respectively, and there are 10 genes located on the scaffolds.

Gene duplication events are essential for the evolution of family members [31]. Segmental duplication and tandem duplication are main driving forces of gene duplication. In order to understand the expansion of the F-box family in poplar, MCScanX was used to analyze their tandem duplication events [32]. In total, 27 tandem duplication genes were found, among which 23 genes were distributed on Chr1, 8, 10, 11, 15 and 17, and the others were located on the scaffolds. All tandem genes on Chr1, 8 and 17 all occurred in adjacent positions. As shown in Figure 4b, 38 genes had segmental duplication; these were distributed on 13 chromosomes (except Chr8, 10, 12, 15, 17 and 19), with six genes on Chr1 and Chr2. These results suggest that segmental duplication may play a critical role in the gene duplication events of the F-box family in poplar.

In addition, the collinearity of F-box genes between poplar and *A. thaliana*, *G. max*, *A. vitifolia* Buch, *O. sativa* and *Z. mays* was comparatively analyzed, and their collinearity maps were drawn by TBtools (Figure 5). The results show that 56 *PtrFBX* genes in total are collinear with other species, and the collinear frequency is high on Chr2. Poplar has the most orthologous genes with *G. max*, as there are 53 *PtrFBX* genes collinear with 58 *G. max* genes; followed by *A. vitifolia* Buch, with 28 genes from *A. vitifolia* Buch being collinear with 31 *PtrFBX* genes. Poplar has fewer orthologous genes with *A. thaliana*, as only 10 *PtrFBX* genes are collinear with 10 *A. thaliana* genes (Appendix A). It is noteworthy that there are four shared genes (*PtrFBX40*, *PtrFBX41*, *PtrFBX48* and *PtrFBX229*) that are collinear with *A. thaliana*, *G. max* and *A. vitifolia* Buch. However, the collinearity relationship was not found between *PtrFBX* with *O. sativa,* and with *Z. mays*. 

In order to determine the selection pressure of the F-box family in poplar during evolution, the value of Ka/Ks was used to characterize the evolutionary ability of genes in genome-wide duplication events (Appendix A). If we considering the case where there is no true value, there are 26 pairs of *PtrFBX* genes. Among these, the Ka/Ks scores of 25 *PtrFBX* genes are less than 1, ranging from 0.089402504 to 0.686712535. The results show that the majority of *PtrFBX* genes are limited in the evolutionary process and are mainly subject to strong purification selection, which is consistent with the report on F-box genes in soybean [9]. On the basis of the results of Ks, T = Ks/2r was used to estimate the approximate time of duplication events, which ranged from 52.78 to 124.61 Mya for segmental duplications and 31.72 to 462.33 Mya for tandem duplications.

### 2.5. Cis-Acting Elements Analysis of PtrFBX Promoter Sequences

*Cis*-acting elements are the key regions of intergenic regulation. Proteins can induce, repress and enhance gene transcription regulation in plants by binding to specific *cis*-acting elements [33]. To screen candidate genes that may be involved in the salt tolerance pathway, the upstream 2000 bp sequences of 245 *PtrFBX* genes were analyzed. Five hormone-related and four stress-related components were found in the promoter regions of *PtrFBX* genes (Appendix A). The hormone-related elements included P-box/TATC-box (response to gibberellin), TGA-element/AuxRR-core (response to auxin), ABRE (response to abscisic acid), the TCA element (response to salicylic acid) and the TGACG motif (response to jasmonic acid). Most *PtrFBX* promoters contained ABRE (172), which is an important regulatory element in response to ABA. Among 156 genes, there were 117 P-box and 39 TATC-box elements that respond to GA. There were two types of auxin-responsive elements, including 72 TGA elements and 23 AuxRR-core elements. As many as 104 genes also contained TCA elements and the TGACG motif. A few stress-related elements such as the WUN motif, the AT-rich element, MBS and LTR were also identified in the *PtrFBX* promoter regions.

### 2.6. Expression Pattern of PtrFBX Genes

After transcriptomic sequencing, 579.53 MB of raw data was obtained in total, and 531.7 MB of clean reads was obtained after filtration. The Q30 base percentage of each sample was above 98.36%, so it could be used for follow-up studies. Using RNA-Seq, we explored the expression patterns of *PtrFBX* genes in the roots, stems and leaves. In this study, 167 *PtrFBX* genes with FPKM ≥ 1 in any tissue were considered to be expressed (Figure 6a). To explore expression patterns of F-box genes under salt stress in poplar, we profiled the mRNA abundance of DEGs across the leaves, stems and roots. As shown in Figure 6c–e, the results showed that 64 *PtrFBX* genes were DEGs in the roots, among which, 23 genes were significantly upregulated and 41 were downregulated. There were 26 DEGs in the stems (13 up- and 13 downregulated) and 17 in the leaves (11 up- and 6 downregulated). Among DEGs, there were eight shared genes between the leaves and roots, nine genes in the comparison of leaves and stems, and 12 shared between the roots and stems (Figure 6b). It should be noted that *PtrFBX1*, *PtrFBX38*, *PtrFBX167* and *PtrFBX168* were DEGs found across three tissues under salt stress, according to the RNA-Seq analysis (Appendix A). In order to verify the accuracy of the RNA-Seq results, 23 DEGs were randomly selected for qRT-PCR (Appendix A). The results showed that the expression trends of most DEGs were consistent between qRT-PCR and RNA-Seq (Figure 7), indicating the accuracy of RNA-Seq. In addition, a spatiotemporal expression analysis of four shared DEGs in the leaves and roots was performed (Appendix A). The results showed that the expression trends of the four genes were similar and were induced, to some extent, by salt stress. In the leaves, the expression levels of the *PtrFBX1*, *PtFBX168* and *PtrFBX228* all reached their peaks at 12 h, and *PtrFBX38* reached the highest level at 24 h, then their expression levels decreased. In the roots, the expression levels of *PtrFBX38*, *PtrFBX168* and *PtrFBX228* continued to increase. *PtrFBX38* and *PtrFBX168* reached the highest level at 36 h and *PtrFBX228* reached the peak at 24 h, while the expression level of *PtrFBX1* did not apparently change.

### 2.7. Co-Expression Network Analysis

In a co-expression network, the genes in the same cluster may display similar functions in a specific pathway, and the unknown genes can be further identified by referring to similar genes in the same pathway. In the study, we selected three DEGs (*PtrFBX1*, *PtrFBX38* and *PtrFBX168*) and one ABA response gene (*PtrFBX228*) to construct co-expression cluster networks by Spearman’s method [34]. Among the four co-expression networks, the network centered on the *PtrFBX228* gene was the largest (788 genes), while the one centered on *PtrFBX1* was the smallest (272 genes) (Appendix A).

In addition, the EggNOG database was used for gene enrichment analysis, and the top 30 GO terms with significant enrichment are displayed in Figure 8. The networks of the four genes are involved in diverse biological processes (Appendix A). Most genes of the *PtrFBX*1 set function in cellular metabolic processes (105), while the genes of the *PtrFBX*38 set are mainly enriched in the membrane (77) and cell periphery (61). The other genes function in stress responses, such as the responses to external stimuli (GO: 0009605), defense responses (GO:0006952), responses to other organisms (GO: 0051707), responses to external biotic stimuli (GO:0043207) and responses to biotic stimuli (GO: 0009607). In particular, the gene sets of *PtrFBX*168 and *PtrFBX*228 share 52 GO catalogs, including responses to stimuli (GO:0050896), biological regulation (GO: 0065007), regulation of biological processes (GO:0050789), responses to chemicals (GO: 0042221), responses to stress (GO: 0006950), responses to abiotic stimuli (GO: 0009628), cellular responses to stimuli (GO: 0051716), responses to acid chemicals (GO: 0051716), responses to hormones (GO: 0009725) and signal transduction (GO: 0007165) and so on.

## 3. Discussion

The F-box family is widely distributed in plants and acts as an important regulator in crucial cellular processes by selectively binding to specific substrates and then degrading the target genes within the ubiquitin/26S proteasome system [18]. F-box genes have been identified in many plant species such as *Arabidopsis*, rice, wheat, chickpea, etc. [12,34,35]. However, most of the F-box’s function remain unknown in plants, especially in poplar. In this study, 245 F-box proteins were identified in poplar, with smaller parameters in the version of the poplar genome database *Populus trichocarpa* 3.0, which was inconsistent with former research [35]. According to the chromosomal distribution, gene structures, conserved motifs and characteristics of the phylogenetic tree of the F-box members, we divided the family into different subfamilies, analyzed the expression patterns of the F-box genes with and without salt stress in poplar using RNA-seq, and validated this by qRT-PCR. Representative genes were subjected to co-expression network analysis, and GO terms enrichment and comprehensive analyses were performed.

The release of the poplar genome makes it possible to conduct a comprehensive analysis of the evolution, structure and function of any family with complete reference sequences and annotations [29]. Many F-box family proteins actively participate in the protein degradation processes in specific pathways, mainly due to their diverse structural domains. Different domains will specifically interact with diverse substrates to perform different biological functions [12]. In this study, many conserved domains, such as LRR, Kelch, FBA, FBD, TUB, FBO (FIST, PAS-Kelch, Actin, etc.), and a combination of unknow domains were found at the C-terminal of F-box proteins. LRR, Kelch, FBA, TUB and FBD are widely present in *Arabidopsis*, wheat and maize, whereas FBU (135 out 245) is considered to be the most abundant domain among different species [12,35]. The second most abundant domain varies greatly among different species. For example, FBD is the second most abundant in rice [12], while it is the FBA domain in poplar and wheat, which may participate in the degradation of target proteins [35].

The expansion of family members through large-scale duplication events is the basis for maintaining the stable existence of a large family in the process of evolution [31]. In order to better adapt to the changing environment, the protein families will expand through duplication events of the genome, such as bZIP [36], LEA [37] , XTH [38] etc.. As it is one of the largest protein families, duplication analysis of the F-box genes showed that 38 out of 245 genes are segmental duplication genes and 27 are tandem genes. The results suggest that segmental duplication contributed more to the expansion of the F-box family, which was also found in wheat [35]. More notably, the pattern of tandem duplication is biased among subgroups. Among the 27 tandem duplication genes, 18 genes belong to the FBU subgroup, which may reveal the direction of replication gene expansion. Ka/Ks values can be used to assess the evolutionary pressure on a gene family. There were 26 of 27 gene pairs with Ka/Ks < 1, indicating that most *PtrFBX* genes may have undergone strong purification selection during evolution. The differentiation of F-box duplication gene pairs can be estimated by the Ks, and the Ks values of segmental duplication and tandem duplication were about 52.78 to 124.61 Mya and 31.72 to 462.33 Mya, respectively. These details suggest that the differentiation of F-box family duplication was not synchronous. We further compared the collinearity relationship between poplar and five species, including three dicotyledons and two monocotyledons. Poplar has a better collinearity relationship with dicotyledons than with monocotyledons, among which *PtrFBX* genes have a best collinearity relationship with *G. max*.

*Cis*-acting elements function as important regulators in the transcription of neighboring genes. This study reported many *cis*-elements, such as ABRE, P-box, TATC-box, TGA-element, MBS, LRT, AT-rich, etc., that were found on the promoter regions of *PtrFBX* genes. These could be divided to two kinds, including hormone response elements and stress-related elements. Among these, ABRE is the most abundant hormone response element in F-box genes (70.2%). *EDL3* has been reported to be ABA-induced and contributes to positive regulation of the root-to-flower transition in the ABA signaling pathway in *Arabidopsis* [39]. TC-repeat is the most abundant stress-related element, which is involved in immunity responses and plays an important role in the development of plant cells [40]. The WUN-motif functions in the process of wounding [41]. MBS is an important binding site of MYB transcription factors and regulates downstream genes through mutual binding under drought stress [42]. LRT is an important response element under low temperatures [43]. These details suggest that poplar’s F-box genes may be involved in plant growth and abiotic stress responses by interacting with other genes.

As an important component of the SCF complex, the F-box family plays a vital role in plant growth, development, signal transduction, aging and senescence [44]. In *Arabidopsis*, F-box genes are vital for maintaining intracellular homeostasis through the auxin, JA and ABA pathways. The phylogenetic analysis of *PtrFBX* genes is helpful for screening similar functional F-box genes from other species. For example, the homologous gene of *PtrFBX196* is *TIR1*, which encodes an auxin receptor and is involved in auxin signal transduction and transcription [45]. The homologous gene of *PtrFBX61*, *AtCOI1*, has been reported to participate in specific protein degradation and to combine with *JAZ1* to form SCF [46]. The homologous gene of *PtrFBX38* is *MAX2*, which is sensitive to ABA and can reduce stomatal closure, thus improving plant tolerance [20].

When plants are subjected to stress stimuli, they make a response through the corresponding life activities, such as signal pathway transduction and gene transcriptional regulation, so as to protect their normal intracellular activities. Transcriptome data provides a new method for exploring expression patterns of the relevant genes. The significant DEGs under the stress treatment may be involved in corresponding pathways. In this study, we explored the expression patterns of *PtrFBX* genes without and with salt stress. There were 82 DEGs in the three tissues in total, including 64 DEGs in the roots, 26 in the stems and 17 in the leaves. *PtrFBX1*, *PtrFBX38* and *PtrFBX168* were significantly upregulated in the three tissues. *PtrFBX167* was only upregulated in the stems and was downregulated in the leaves and roots. To better understand the function of these DEGs, we identified their homologous genes in the TAIR database (https://www.arabidopsis.org/ (accessed on 10 August 2021)). For example, the homologous gene of *PtrFBX12*, *PtrFBX52* and *PtrFBX54* is *AT1G273401*, which can enhance tolerance to drought and salt in *Arabidopsis* [47]. *At2G47900.1*, which is homologous to *PtrFBX228*, can enhance ABA biosynthesis to promote bacterial susceptibility [48]. The homologous gene of *PtrFBX143* is *AT4G24210.1*, which acts as a negative regulator and can antagonize the GA signal in the GA pathway [49]. Similarly, *AT1G68050.1*, the homolog of *PtrFBX121*, has been reported to negatively regulate DELLA proteins to promote flowering during long daylight cycles [50]. It has been reported that *AT1G15670.1* is a negative regulator of cytokinin, and is the homologous gene of *PtrFBX1* [51].

In the present study, we also conducted a co-expression network analysis of four interesting genes including three DEGs and one ABA-related response gene. These genes were annotated to be involved in abiotic stress responses and complex physiological processes. The gene networks of both *PtrFBX*168 and *PtrFBX*228 are significantly enriched in the responses to stimuli, biological regulation and responses to stress. The *PtrFBX1* gene sets mainly focus on cell metabolic processes and responses to oxygen-containing compounds, while the *PtrFBX*38 gene sets cover the cell periphery and biological processes involved in interspecies interactions between organisms. The results indicate that different regulators operate complex protein regulatory networks.

## 4. Materials and Methods

### 4.1. Identification of F-box Proteins and GO Functional Annotation in Populus Trichocarpa

The sequences of F-box proteins of *Populus trichocarpa* 3.0 were downloaded from the Phytozome database (https://Phytozome.jgi.doe.gov/pz/portal.html (accessed on 20 June 2021)). The representative Hidden Markov model (HMM) profiles of F-box (PF00646), F-box-like (PF12937), F-box-like 2 (PF13013), FBA (PF04300), FBA_1 (PF07734), FBA_2 (PF07735), FBA_3 (PF08268) and FBD (PF08387) were obtained from Pfam database (http://pfam.xfam.org/ (accessed on 22 June 2021)). The HMM Search program with a threshold of ≤1 × 10^−10^ (https://www.ebi.ac.uk/Tools/hmmer/search/hmmsearch (accessed on 20 June 2021)) was applied to identify the candidate proteins. The SMART website (http://smart.embl.de/ (accessed on 22 June 2021)) and Pfam database were used to verify putative proteins. In addition, the EggNOG database (http://eggnog-mapper.embl.de/ (accessed on 25 June 2021)) was used for functional prediction of the *PtrFBX* genes and for gene ontology (GO) enrichment analysis. The GO enrichment results were displayed by TBtools [52].

### 4.2. Multiple Sequence Alignment and Phylogenetic Analysis

To explore the evolutionary relationships of the F-box family, 245 *PtrFBX* proteins with full-length sequences were obtained, and ClustalW software was used for multiple sequence alignment [53]. MEGA-X was used for construction of the phylogenetic tree by the neighbor-joining (NJ) method [54], and the parameters were set as follows: 1000 iterations of the bootstrap values; JTT (Jones–Taylor–Thornton) + G (gamma distributed) model; optional partial deletion of 95% of the threshold. Evolview online software (http://www.evolgenius.info/evolview/#/treeview (accessed on 5 July 2021)) was used for visualization.

### 4.3. Gene Sequence Analysis

SMART and the Pfam database were used to identify the conserved F-box domain located at the N-terminal of putative F-box proteins in poplar. According to multiple sequence alignment, the F-box domain logo was generated by Weblog (http://weblogo.berkeley.edu/logo.cgi (accessed on 5 July 2021)). On the basis of the highly conserved domains at the C-terminal, F-box members can be further divided into different subfamilies. Furthermore, MEME (http://meme-suite.org/tools/meme (accessed on 5 July 2021)) was used to predict the conserved motifs, and the following parameters were set: Classic mode; zero or one occurrence per sequence (zoops); number of motifs: 20; optimum motif width, ≥6 and ≤50.

### 4.4. Analysis of Gene Locations, Duplication Events and Ka/Ks

The Multiple Collinearity Scan toolkit was used for analyzing the duplication events of the F-box family in poplar [32]. According to the GFF3 file of *P. trichocarpa 3.0* in the Phytozome database, we obtained the detailed chromosome locations of each *PtrFBX* gene. On the basis of the tandem duplication calculated by MCScan, we used TBtools to map the corresponding chromosomes of the *PtrFBX* genes. The Advanced Circos package was used to visualize the duplication relationships of *PtrFBX* gene pairs [52]. To further verify the collinearity of the F-box family among different species, the protein sequences and GFF3 annotation files of the F-box family of *Arabidopsis thaliana*, *Glycine max*, *Anemone vitifolia* Buch, *Oryza sativa* and *Zea mays* were also downloaded from the Phytozome database. The Dual Synteny Plotter package was used to calculate and visualize the collinear relationships of homologous proteins between *P. trichocarpa* and other species. The Ka/Ks calculator was used to verify the occurrence of duplication events between duplication gene pairs. The values of Ka (nonsynonymous substitution rate), Ks (synonymous substitution rate) and Ka/Ks were calculated, and Ks was used to calculate the approximate duplication time via the formula T = Ks/2r, and the value of r was 1.5 × 10^−8^ substitutions per site per year for poplar [55].

### 4.5. Cis-Element Prediction of F-box Promoters in Poplar

The upstream 2000 bp promoter sequences of the *PtrFBX* genes were obtained from the Phytozome database. The *cis*-elements were predicted by the online tool PlantCARE (http://bioinformatics.psb.ugent.be/webtools/plantcare/html/ (accessed on 15 July 2021)) and visualized by TBtools software.

### 4.6. Plant Materials, Treatment and Transcriptome Analysis

The di-haploid *Populus simonii × Populus nigra* was used for research in the State Key Laboratory of Tree Genetics and Breeding, Northeast Forestry University, Harbin, China. The *P. simonii × P. nigra* seedlings were first cultured on half-strength MS culture medium for 20 days under greenhouse conditions, 25 ℃ with a 16:8 h light–dark cycle. Healthy seedlings at a similar growth stage were transplanted to pots filled with soil: vermiculite: perlite = 3:2:1 for 1 month. The soil-cultured seedlings were then treated with a 150 mM NaCl solution for 12 h, 24 h and 36 h, with water as a control. Twelve samples for 0 h and 24 h, including the leaves, stems and roots, with two biological replicates, were paired-end sequenced on the Illumina HiSeq 2500 platform with a 10× sequencing depth. Trimmomatic v0.30 and FastQC v0.10.1 were used to sequence the raw data for quality filtering and analysis [56,57]. Hisat v0.1.6 software was used to map the resulting clean data to the reference genome of *Populus trichocarpa* [58]. The roots, stems and leaves of samples without the salt treatment were used as the control group, and the roots, stems and leaves of the 150 mM salt treatment were used as the experimental group. The mRNA abundance of all *PtrFBX* genes is presented as FPKM (fragments per kilobase million). The DESeq2 package in R was used for screening differentially expressed genes (DEGs) with two standards: false discovery rate (FDR) ≤0.05 and log2 fold change (FC) ≥1 [59]. The heatmaps of DEGs were generated with TBtools software. A Venn diagram of the DEGs was constructed by VEENY 2.1.0 (https://bioinfogp.cnb.csic.es/tools/venny/index.html (accessed on 18 July 2021)).

### 4.7. Validation of DEGs by qRT-PCR

In order to verify the accuracy of the RNA-Seq, the relative expression levels of 23 genes were verified by qRT-PCR. The *Populus Actin* gene was used as an internal reference [60]. The 2^−ΔΔCt^ method was used to determine the relative expression levels of each gene, based on the expression level of each gene without the salt treatment. The sequences of the qRT-PCR primers are listed in Appendix A.

### 4.8. Co-Expression Network and GO Annotation

To further explore genes that may respond to salt stress, we used 12 samples of three tissues (leaves, roots, stems), including 6 untreated and 6 salt-stressed samples, and conducted a network analysis of four interesting genes by Spearman’s method [34]. Cytoscape was used to draw the co-expression networks of the four genes [61]. The genes with a *p*-value of ≤0.05 and a correlation coefficient of ≥0.8 were considered to be co-expressed genes. The EggNOG database was used for gene ontology annotation of the gene sets, TBtools was used to display the top 30 GO enrichment figures, and the correlation *p*-value of each annotation pathway was corrected by the Benjamin–Hochberg method [62].

## 5. Conclusions

In this study, 245 F-box family members in total were identified in poplar, which were distributed on 19 chromosomes and six scaffolds. On the basis of the C-terminal domains, the F-box family was divided into 16 groups in poplar. The phylogenetic analysis indicated that F-box proteins in the same group shared similar structures and motifs. Segmental duplication was main driving force for the expansion of F-box family members in poplar, and the F-box family has been under strong purification selection in the process of evolution. The promoter regions of *PtrFBX* genes mainly contain two kinds of *cis*-elements, which are related to hormone and stress responses. In addition, the expression patterns of F-box genes in the roots, stems and leaves of poplar were profiled by RNA-Seq under salt treatment. There were 82 DEGs in the three tissues in total, including 64 DEGs in the roots, 26 in the stems and 17 in the leaves. Co-expression network analysis of four representative *PtrFBX* genes showed that their co-expressed genes were involved in abiotic stress responses and complex physiological processes. The comprehensive analysis of F-box genes in poplar will provide new insights for understanding the evolution of the F-box family and their functions in response to abiotic stresses.

## Figures and Tables

**Figure 1 ijms-23-10934-f001:**
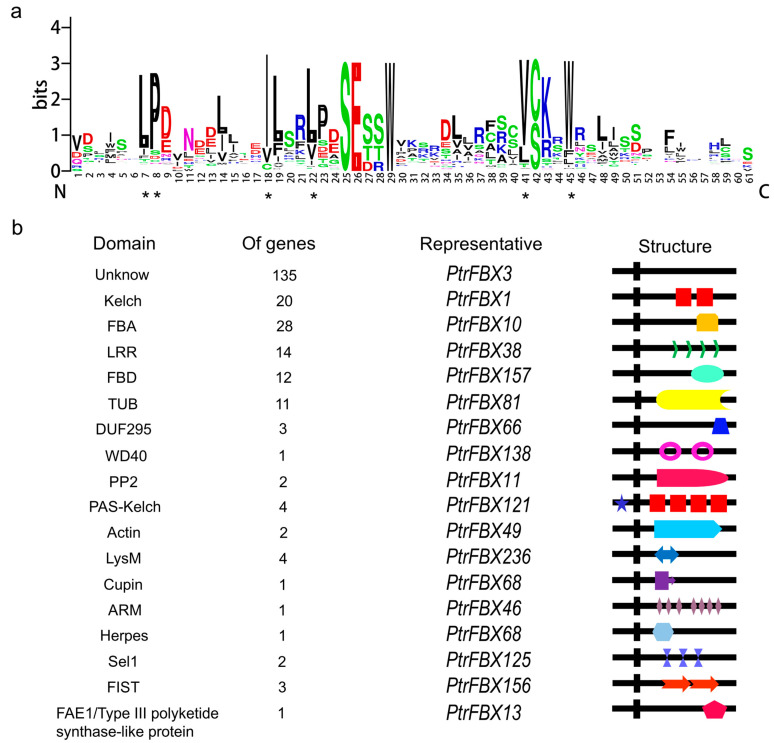
Domain analysis of the F-box proteins in poplar. (**a**) Sequence logo of the conserved F-box domain of the 245 F-box proteins. The x-axis represents the amino acid residues positions in the F-box domain. The y-axis represents the number of times that the amino acid appears in a particular position. The height of a single letter represents the relative frequency of the corresponding amino acid at that position. ***** represents the conserved amino acid residues of F-box domain. (**b**) The conserved domain names represent on the left, gene numbers and representative gene containing corresponding domain are in the middle, domain features are on the right. Black vertical box represents a F-box domain, boxes with different colors represent corresponding domains of C terminal.

**Figure 2 ijms-23-10934-f002:**
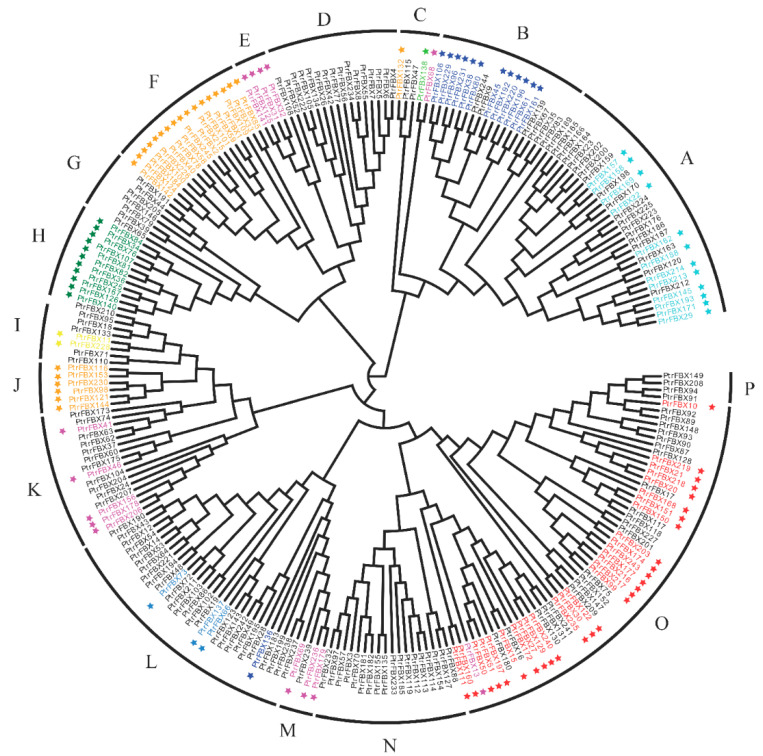
Phylogenetic tree of the F-box proteins in poplar. The phylogenetic tree using 245 full-length protein sequences was generated by MEGA-X by neighbor-joining (NJ) with the JTT + G model. All F-box proteins were divided into 16 groups (A–P). Different star colors represent different subgroups: Orange, FBK subgroup; green, FBT; dark blue, FBL; cyan, FBD; yellow, FBP; red, FBA; Dodger blue, FBF; lime, FBW; magenta, FBO.

**Figure 3 ijms-23-10934-f003:**
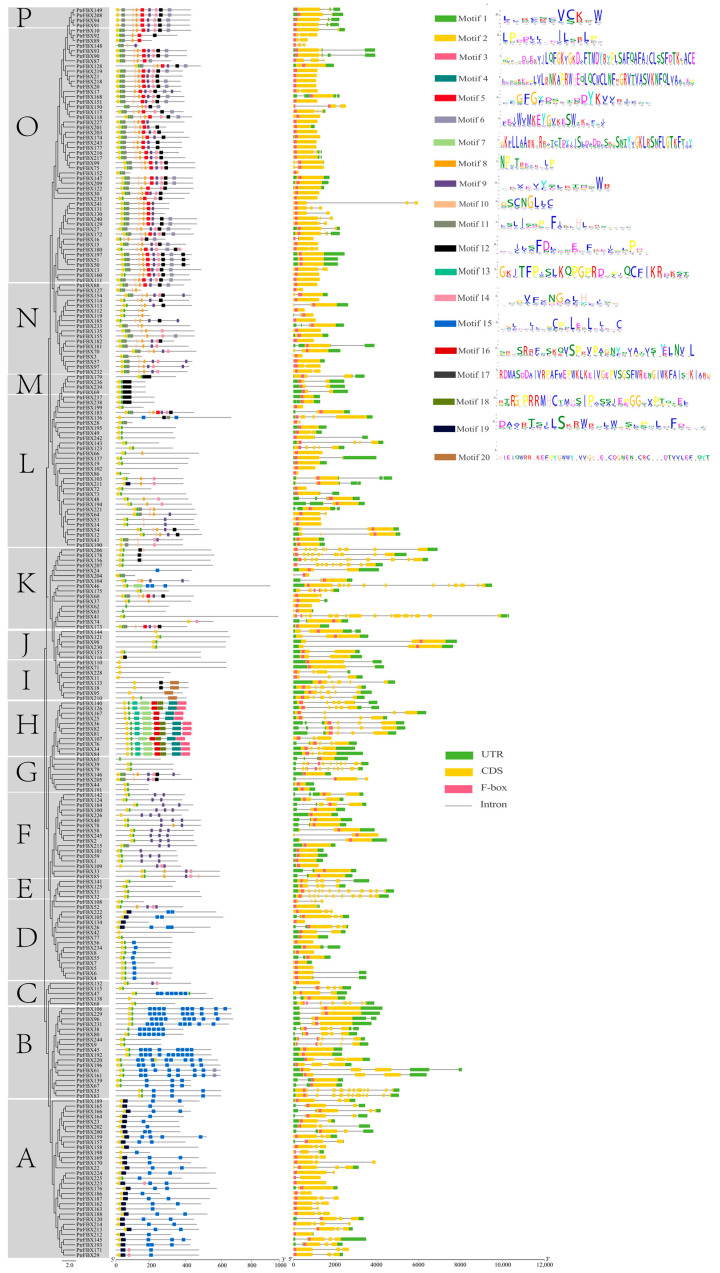
Structure and motif analysis of *PtrFBX* proteins. The subgroups (A–P) and motif analysis are on the left. The clustering is displayed according to the results of phylogenetic tree. Different colored boxes represent different motifs. The motif sequences are on the right. Gene structures are in the middle, where the green boxes represent untranslated regions, the yellow boxes represent exons and the grey lines represent intron regions.

**Figure 4 ijms-23-10934-f004:**
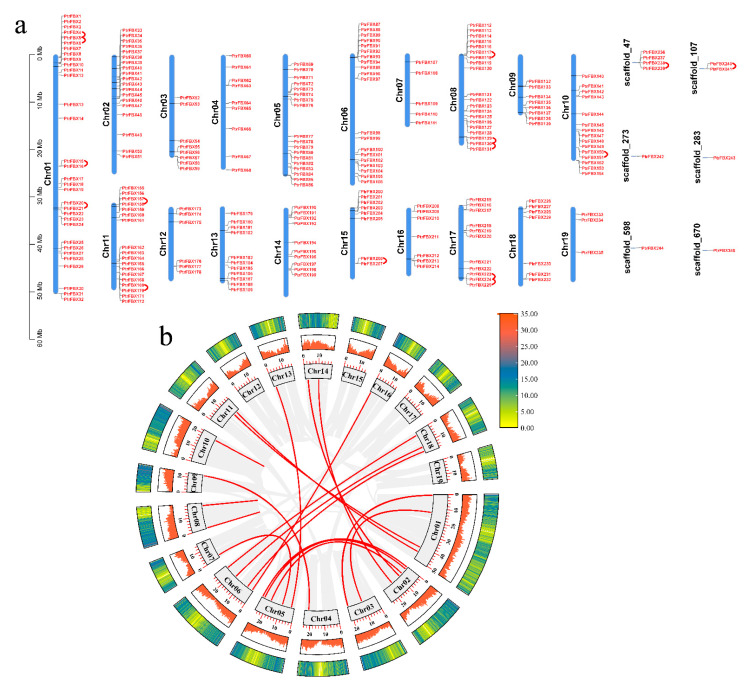
Chromosome location and duplication events of the F-box genes in poplar. (**a**) The distribution of 245 *PtrFBX* genes on poplar’s chromosomes and scaffolds. Chr 01–19 represents chromosomes 1–19. The chromosomes are drawn to scale and the chromosome numbers are indicated on the left. Gene names are to the right of the chromosomes. The red lines represent tandem duplication gene pairs. (**b**) Circle map of the duplication gene pairs of the *PtrFBX* genes. The heatmap and the histograms in rectangles represent the gene density on the chromosomes. The red lines represent duplication gene pairs of *PtrFBX* genes, while the gray lines indicate collinear blocks of all poplar genes.

**Figure 5 ijms-23-10934-f005:**
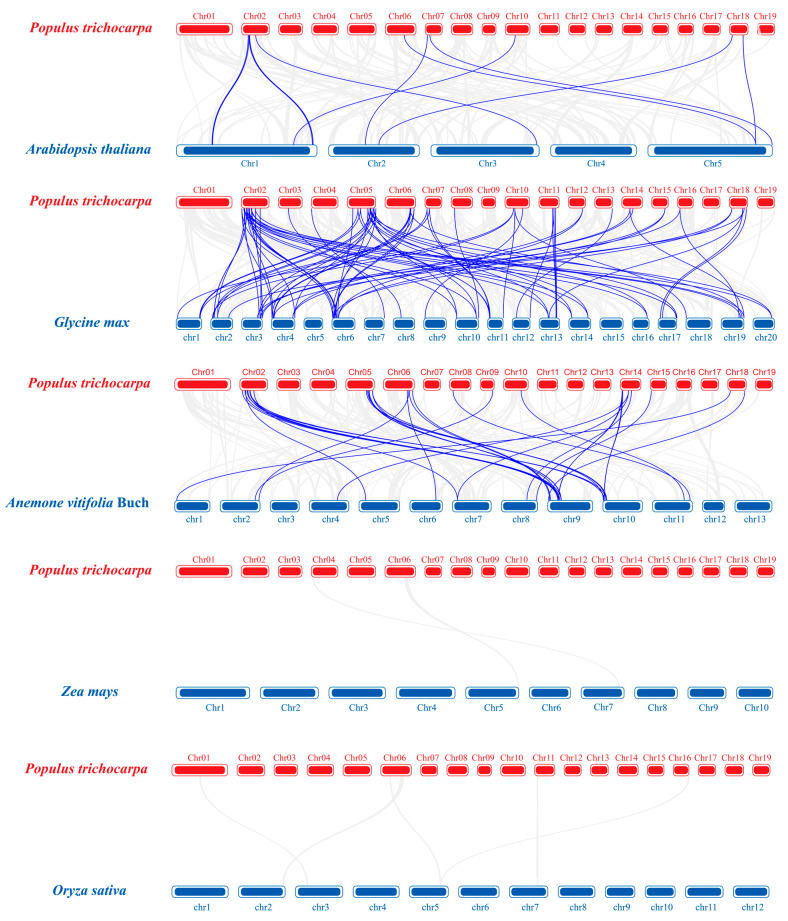
Synteny analysis of F-box genes between poplar and other species. The gray lines represent orthologous gene blocks between poplar and other species. The blue lines display the orthologous F-box gene between *p**opulus trichocarpa* and other species. The red and blue thick lines in rectangles represent the corresponding chromosomes of *p. trichocarpa* and other species, respectively.

**Figure 6 ijms-23-10934-f006:**
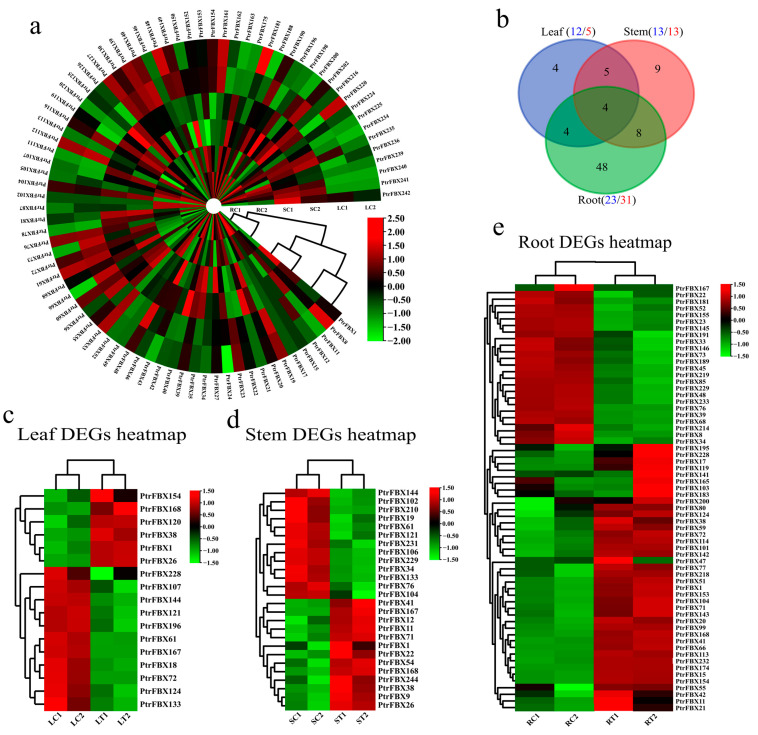
Expression patterns of *PtrFBX* genes. All heatmap values are scaled by log2^FPKM^. (**a**) Heatmap of the roots, leaves and stems of *Populus simonii × Populus nigra* without salt treatment. (**b**) Venn diagram of DEGs across the three tissues. Blue, pink and green circles denote DEGs of the leaves, stems and roots, respectively. (**c**–**e**) Heatmap of DEGs in the leaves, stems and roots before and after salt stress. S, L and R denote the stems, leaves and roots, respectively. T and C represent with and without salt treatment, respectively. Red and green boxes denote relative upregulation and downregulation of expression, respectively.

**Figure 7 ijms-23-10934-f007:**
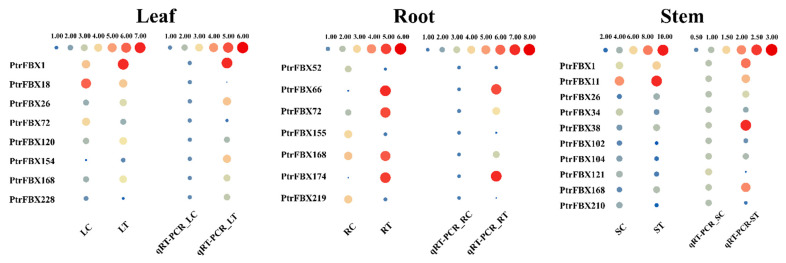
Relative expression level of DEGs under salt stress based on RNA-Seq and qRT-PCR across the three tissues of *Populus simonii* × *Populus nigra*. The gene relative expression was calculated, based on the expression level of each gene without salt treatment. Blue and red circles represent relatively high and low expression, respectively. The size of the circle represents the relative expression level. The larger the circle, The larger the circle, the higher the relative expression level of the gene.

**Figure 8 ijms-23-10934-f008:**
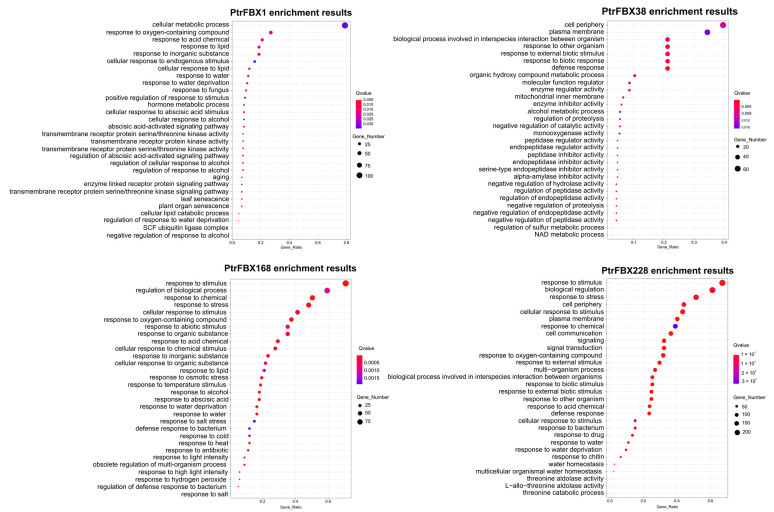
The top 30 significant GO enrichment annotations of the four representative gene sets. The *Q* value is represented by color (red and purple): the darker the red, the smaller the *Q* value; on the contrary, the closer to purple, the larger the value. The size of the circle represents the number of enriched genes in a certain pathway. The larger the circle, the more genes are enriched in that pathway.

## Data Availability

All data generated or analyzed during this study are included in this published article and information files. Informed consent was obtained from all subjects involved in the study. The raw sequencing data used during this study have been deposited in the NCBI’s SRA with the accession number SRP267437 (https://trace.ncbi.nlm.nih.gov/Traces/sra/?study=SRP267437 (accessed on 17 July 2021)).

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
