# Peer review of "Genome-Wide Identification and Expression Patterns of the F-box Family in Poplar under Salt Stress"

_ijms, 2022, doi:10.3390/ijms231810934_

Round 1

Reviewer 1 Report

The manuscript systematically analyzed F-box family in polar and explored its expression pattern under salt stress. Overall, it is a well-written paper and easy to follow. However, there are minor issues that the authors need to address before the manuscript can be considered for publication. 

1. When you retrieve F-box family of Arabidopsis thaliana, Glycine max, Anemone vitifolia Buch, Oryza sativa and Zea mays, did you use SMART website and Pfam database to verify putative proteins? Besides, how did you obtain F-box proteins from these species? Based on annotated names in GFF3 annotation files or conduct similarity searching? 

2. To my understanding, all of the F-box sequences used in the present study are obtained from the predict protein database or protein database based on primary transcripts, and the authors did not conduct any homology analysis at the genome level. As a result, some potential F-box genes might be omitted from the genome. I suggest the author carefully present some related sentences, especially in cross-species synteny analysis section. 

3. Fig 1. What do the structures and colors mean in domain structure? Related information is necessary.

4. Did you find any correlation among phylogenetic results, cis-acting elements and transcriptomes. Consider adding some discussion to connect difference parts of this manuscript.

Author Response

Author's Reply to the Review Report (Reviewer 1)

The manuscript systematically analyzed F-box family in polar and explored its expression pattern under salt stress. Overall, it is a well-written paper and easy to follow. However, there are minor issues that the authors need to address before the manuscript can be considered for publication.

  1. When you retrieve F-box family of Arabidopsis thaliana, Glycine max, Anemone vitifolia Buch, Oryza sativa and Zea mays, did you use SMART website and Pfam database to verify putative proteins? Besides, how did you obtain F-box proteins from these species? Based on annotated names in GFF3 annotation files or conduct similarity searching?

Thank you for your valuable advice. We aligned the protein similarity of poplar and other species according to GFF3 and genome, and determine collinear genes between the whole genomes of two species. Then, the identified poplar F-box genes are matched with the collinear genes, and the selected candidate genes are further identified by the F-box domain through SMART and Pfam databases. Based on the above screening methods, the determined candidate genes were identified as genuine collinear genes.

  1. To my understanding, all of the F-box sequences used in the present study are obtained from the predict protein database or protein database based on primary transcripts, and the authors did not conduct any homology analysis at the genome level. As a result, some potential F-box genes might be omitted from the genome. I suggest the author carefully present some related sentences, especially in cross-species synteny analysis section.

Thank you for your valuable advice. We have added some details in cross-species synteny analysis section.

  1. Fig 1. What do the structures and colors mean in domain structure? Related information is necessary.

Thank you for your valuable advice. We have added some details in Fig 1. annotation.

  1. Did you find any correlation among phylogenetic results, cis-acting elements and transcriptomes. Consider adding some discussion to connect difference parts of this manuscript.

Thank you for your valuable advice. We found that DEGs are unevenly distributed in other subgroups (except M and P subgroups), and the vast majority of the promoter sequences of the DEGs contain ABRE elements, indicating that the F-box family may be affected by ABA or participated in ABA pathway. It has been reported that The FBA motif-containing protein AFBA1 is a novel positive regulator of Arabidopsis ABA response. Transcriptome analysis revealed Arabidopsis F-BOX STRESS INDUCED 1 as a regulator of jasmonic acid and abscisic acid stress gene expression. The F-box gene FOA2 regulates GA- and ABA-mediated seed germination in Arabidopsis. We have added some details in discussion.

Reviewer 2 Report

1)In Figure 1b, where are the detailed annotations of each domain?

2)Figure 2 what is the basis for grouping various F-box proteins? Is amino acid sequence homology, domain difference, or both?

3)Figure 3 is not clear at all. Is there a clearer display?

4)What do the three broken red lines in Figure 4B mean? No homologous genes in the genome?

5)In Fig. 5, the synteny analysis of F-box protein of poplar between corn or rice is not listed in the figure.

6)The results of some RT-PCR in Fig. 7 are too large and unreliable. Why not do it a few more times? In addition, the results are better presented in the form of heat maps.

Author Response

Author's Reply to the Review Report (Reviewer 2)

Comments and Suggestions for Authors

1)In Figure 1b, where are the detailed annotations of each domain?

Thank you for your valuable advice. We have added some details in Fig. 1 annotation.

2)Figure 2 what is the basis for grouping various F-box proteins? Is amino acid sequence homology, domain difference, or both?

Thank you for your valuable advice. We firstly classified according to the similarity between amino acids in the MEGA phylogenetic tree, and divided most of the groups with similar domains and motifs into the same group.

3)Figure 3 is not clear at all. Is there a clearer display?

Thank you for your valuable advice. We have modified Fig. 3.

4)What do the three broken red lines in Figure 4B mean? No homologous genes in the genome?

Thank you for your valuable advice. Segmental duplication and tandem duplication are main driving force of gene duplication. We also count tandem duplication in the circle map, and the three red lines represent the four tandem duplication gene pairs present on Chr 8 and 10.

5)In Fig. 5, the synteny analysis of F-box protein of poplar between corn or rice is not listed in the figure.

Thank you for your valuable advice. We found PtrFBX genes had no collinear genes with O. sativa and Z. mays in my study.

6)The results of some RT-PCR in Fig. 7 are too large and unreliable. Why not do it a few more times? In addition, the results are better presented in the form of heat maps.

Thank you for your valuable advice. We performed technical replicates of the real-time quantitative data, corrected for biased results, and presented the experimental results in the form of a heatmap in Fig. 7.

Reviewer 3 Report

The work is devoted to the study of genes of the F-box family in poplar. This family is known to play an important role in the growth, development, and response of many plant species to stress. The authors, using high concentrations of NaCl on poplar plants as a stress factor, managed to identify 245 PtrFBX proteins and, based on their C-terminal conserved domains, construct a phylogenetic tree divided into 16 groups. These proteins were located among 19 chromosomes and 6 scaffolds. Using bioinformatics methods, the authors investigated the structure, evolution, and expression patterns of poplar F-box genes, which provided the key to understanding the molecular function of F-box family members and screening for PtrFBX salt tolerance genes.

There are no comments on the text, meaning and discussion of the results of the work.

There are comments on References

I ask the authors to carefully check once again the list of used literature, in particular NN 36, 42, 43. The names of the journals are not indicated here.

Author Response

There are no comments on the text, meaning and discussion of the results of the work.

There are comments on References

I ask the authors to carefully check once again the list of used literature, in particular NN 36, 42, 43. The names of the journals are not indicated here.

Thank you for your valuable advice. We have modified references carefully.

Round 2

Reviewer 2 Report

I have no questions.